# Multidecadal Trend Analysis of Armenian Mountainous Grassland and Its Relationship to Climate Change Using Multi-Sensor NDVI Time-Series

**Vahagn Muradyan** [1,*]**, Shushanik Asmaryan** [1]🆔**, Grigor Ayvazyan** [1] **and Fabio Dell'Acqua** [2,*]🆔

[1]  Centre for Ecological-Noosphere Studies, National Academy of Sciences, Abovyan Street 68, Yerevan 0025, Armenia

[2]  Department of Electrical, Computer and Biomedical Engineering, University of Pavia, Via A. Ferrata 5, 27100 Pavia, Italy

*  Correspondence: vahagn.muradyan@cens.am (V.M.); fabio.dellacqua@unipv.it (F.D.)

**Abstract:** This paper presents a comprehensive analysis of links between satellite-measured vegetation vigor and climate variables in Armenian mountain grassland ecosystems in the years 1984–2018. NDVI is derived from MODIS and LANDSAT data, temperature and precipitation data are from meteorological stations. Two study sites were selected, representing arid and semi-arid grassland vegetation types, respectively. Various trend estimators including Mann–Kendall (MK) and derivatives were combined for vegetation change analysis at different time scales. Results suggest that temperature and precipitation had negative and positive impacts on vegetation growth, respectively, in both areas. NDVI-to-precipitation correlation was significant but with an apparent time-lag effect that was further investigated. No significant general changes were observed in vegetation along the observed period. Further comparisons between results from corrected and uncorrected data led us to conclude that MODIS and LANDSAT data with BRDF, topographic and atmospheric corrections applied are best suited for analyzing relationships between NDVI and climatic factors for the 2000–2018 period in grassland at a very local scale; however, in the absence of correction tools and information, uncorrected data can still provide meaningful results. Future refinements will include removal of anthropogenic impact, and deeper investigation of time-lag effects of climatic factors on vegetation dynamics.

**Keywords:** grassland; time-lag effects; BRDF; radiometric corrections; Armenian mountainous environment

## 1. Introduction

Surface vegetation is one of the most important components of ecosystems on Earth, playing a key role in regulating carbon balance and climate stability [1–3]. Vegetation is highly sensitive to climate change, particularly in mountain regions [4]. Mountain ecosystems are considered to be among those most severely and rapidly impacted by climate change [3–6]. Thus, over the recent decades, monitoring vegetation dynamics and relationships with climate data has been recognized as a hot issue in global and regional change studies, especially since satellite data started becoming increasingly available [7–10]. In order to comprehensively understand the impact of climatic factors on vegetation dynamics, it is necessary to perform location-specific case studies on correlations between vegetation and climate factors in different geographical regions [6,11–14].

Remote sensing technologies as an alternative to time-consuming and labor-intensive field experimentation offer effective approaches that have been widely used to monitor surface vegetation dynamics in context of climate change [5,13–16]. Remote sensing products such as the normalized difference vegetation index (NDVI) are widely used to monitor the dynamics of vegetation in ecosystems and can be used as a proxy of vegetation response to

climate change [8,17–33]. In recent years, time-series NDVI datasets have been adopted to monitor vegetation dynamics and explore the relationship between NDVI and climate factors in different geographic regions [26,33–45] including both regional [12,46,47] and global scales [30,48–52].

The literature analysis shows that most of these studies were conducted on NDVIs derived from satellite images at coarse spatial resolutions (NOAA/AVHRR, SPOT/VGT and MODIS) [9,10,12,31,35,38,41,43,49–56], which are inappropriate for resolving local spatial variations. Satellite data at high temporal resolutions are informative, but the coarse spatial resolution can suppress important changes at a very local scale. Moreover, MODIS and SPOT/VGT data only cover a limited time period (2000–2018) and may not be able to fully capture potential climate change impacts. On the contrary, the spatial resolution of the LANDSAT sensors is significantly finer and can cover a much longer time period (1984–2018), which facilitates detection of vegetation response to climate changes at a local scale [19,27,56–59]. Most of previous studies widely used LANDSAT time series, which were focused on change detection and monitoring of vegetation [48,59–64]. However, use of the LANDSAT time series to research on the relationships between vegetation and climate factors is limited [65,66].

Analyses of previous studies demonstrated that there was a similar performance of MODIS and LANDSAT data [59,64,67,68]. Therefore, we can use the low spatial resolution MODIS data to study relationships between NDVI and climate factors [69].

Although the individual LANDSAT sensors have changed through time, the spectral characteristics from LANDSAT 4–8 are reasonably comparable and support the generation of dense time-series spectral information [27,70]. However, LANDSAT satellite data received from different sensors need to be pre-processed before drawing final conclusions about vegetation dynamics [71,72].

To obtain more accurate results, usually LANDSAT and MODIS data are pre-processed using a number of methods: cloud masking, atmospheric correction, topographic correction, and bidirectional reflectance distribution function (BRDF) correction [61,66,73,74]. BRDF correction with atmospheric correction is important for creating long-term time series of satellite data and allowing comparison between measurements of NDVI from different sensors and times [75–78], including both LANDSAT TM/ETM/OLI [59,73,74] and MODIS [79,80].

Satellite surface reflectance (SR) values are adjusted to a nadir view and local solar observation geometry to provide nadir BRDF-adjusted SR data [81]. The BRDF correction method is used to correct differences in the field of view angles among satellites and the data received from different times of the same sensor, as the satellite orbit changes over time, and it can have an impact on NDVI values [82,83]. In addition to the BRDF correction method, atmospheric and topographic corrections are also important. Atmospheric elements such as water vapor, total ozone column, and aerosol optical depth affect the accuracy of satellite-based NDVI data [84]. The differences in the NDVI values occur depending on the atmospheric correction method [85]. Topographic correction of satellite images over mountainous areas is very important [86], especially when the data are to be used for monitoring of surface vegetation [87,88]. To our knowledge, only a few studies have reported the use of the BRDF, atmospheric, and topographic-corrected NDVI data with high spatial resolution in studying the impact of climate change on vegetation cover.

Many previous studies have reported strong correlations between NDVI and main climatic factors of precipitation and temperature [89–93]. Furthermore, the relationships between climatic factors and NDVI/vegetation are different in various geographical regions and types of land cover [41,88,94]. However, as a general pattern, precipitation is the climatic parameter less correlated with the NDVI. On the contrary, the most correlated climatic parameter is temperature [95].

In recent years, several studies have verified that the response of vegetation/NDVI to climatic factors feature obvious time-lag effects [22,37,46,96–103], which indicates that vegetation growth may be primarily affected by past climate conditions. Furthermore,

time-lag duration varied among climatic zones and land-cover types, generally the lag time of vegetation in arid areas is longer than that in humid areas [25,46]. The same type of vegetation has different time-lag effects by different climatic factors, and different vegetation types respond differently to the same climatic factor [46,96]. In the scientific literature, most previous studies on time-lag responses were based on monthly or mean growing season NDVI data [6,104,105], which is not conducive to determining the lag. Therefore, in order to understand the dynamic of mountain ecosystems and their relationships with climatic factors, it is preferable to use 10-day time-lag data [106]. The information of the time-lag effects of climate change on NDVI is necessary to discover the mechanisms underlying climate vegetation relationships. Furthermore, it is necessary to take the time lag into consideration in grassland management strategies [107].

Over the last decades, a number of studies have been implemented on climate change in Armenia and the Caucasus. Most of these researches concerned the spatiotemporal trends of climate factors [21,108–110]. Other studies investigated the impact of climate change on the ecosystems of Armenia [111–113] and the Caucasus [58]. The previous study on the territory of Armenia concerned variations of NDVI and climatic factors, their relationships, and time-lag effects using SPOT/VGT data with the limited time series of 1998–2013 and a low spatial resolution. However, the time-lag effects are still poorly understood due to the focus on simultaneous climate conditions for the mountain ecosystem of Armenia. The studies of relationships between climatic factors and NDVI data derived from satellite images with high spatial resolution and with BRDF, atmospheric, topographic corrections in the local level in mountain regions is limited [66,88]. In order to understand the complex impact of climatic factors on surface vegetation dynamics, it is necessary to perform the study on a local scale.

Our study focuses on the grassland ecosystems, which is one of the most sensitive land cover types in mountain regions. Research on grassland carbon stocks in arid and semi-arid regions has attracted a great deal of attention in recent years [114,115]. Grasslands are one of the most prevalent and widespread land cover vegetation types, covering more than a quarter of the global land area [116]. In mountainous regions, rural natural grassland ecosystems are highly sensitive to natural changes, and for that reason it is essential to monitor their dynamics in the context of climate change [16,117,118]. Remote sensing technologies are an effective tool for monitoring grassland ecosystems from local to global scales [117,119]. Grassland ecosystem dynamics reflect the global and regional scale of natural processes and local-scale anthropogenic changes [120]. However, two grassland areas with minimal anthropogenic impact were selected [121] in order to obtain the most realistic results for the relationships between climatic factors and natural vegetation changes.

The goal of this research was to study the time-lag effects of the vegetation responses to climate variables in the mountain grassland ecosystems during 1984–2018 using MODIS- and LANDSAT-derived NDVI data with BRDF, with topographic and atmospheric corrections applied, and temperature and precipitation data from meteorological stations and the Google Earth Engine (GEE) cloud platform [122]. GEE is a cloud-based geospatial analysis platform for scientific analysis and visualization of geospatial datasets; it enables processing of satellite imagery to detect changes, which has been widely used in similar studies in recent years [34,69,123].

## 2. Materials and Methods

### 2.1. Study Area

The Sisian study site (1600–2000 m above sea level, 45°59′57.95″ E, 39°31′57.55″ N) and the Meghri study site (–600–1000 m above sea level, 46°16′23.77″ E, 38°54′44.10″ N), two typical arid and semi-arid grasslands on the Syunk administrative region near to meteorological stations, were selected as the study areas (Figure 1). Sisian and Meghri study sites are covered areas of around 200 ha, dominated by steppe grass and 400 ha, dominated by Mediterranean xerophytic grassland [124], respectively. The Syunik administrative region covers an area of some 4506 km$^2$ in the southeast of Armenia and is characterized

by specific natural and economic conditions. This region has a dry climate with an average annual temperature of 13.8 °C. The spatial distribution of annual precipitation is quite irregular, and it may be hypothesized that such irregular distribution may have a role in reducing correlation with NDVI. The growing season period lasts from April to October [125].

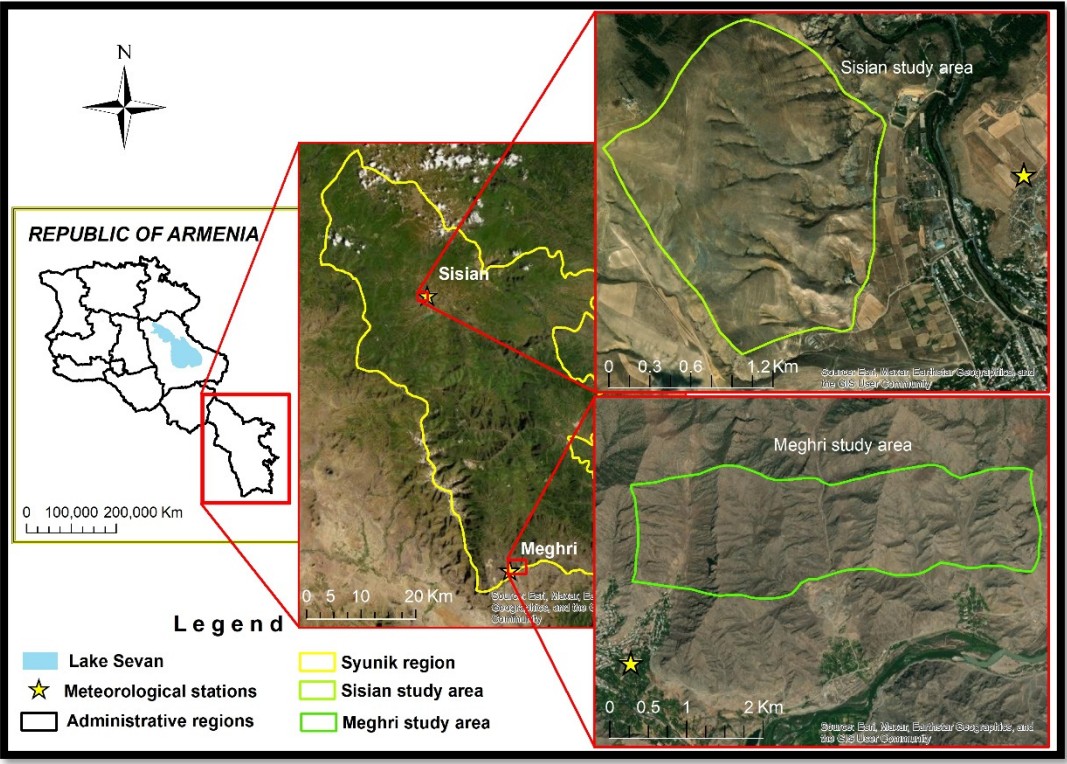

**Figure 1.** Study area.

### 2.2. Data Sources and Processing

#### 2.2.1. Data Acquisition

In this paper, satellite and climatic data were used. Both datasets cover the period 1984–2018, whose length is defined by several factors. First, statistical analysis of trends in this context requires investigations over three decades at least [47]; second, satellite images with sufficiently high resolution must be available; and finally, discovery of climatic trends needs long time spans. It was considered that climate warming generally intensified after the 1980–1990 decade both globally [126] and nationally [108–110].

#### 2.2.2. Remote Sensing Data

The satellite data LANDSAT MSS/TM/ETM+/OLI SR Tier 1 and MODIS Nadir BRDF-Adjusted Reflectance, MODIS SR were accessed and processed for the intended 35 years through the Google Earth Engine (GEE) platform [122] using a JavaScript application programming interface (API). The MCD43A4 V6 Nadir Bidirectional Reflectance Distribution Function Adjusted Reflectance (NBAR) product (MODIS/006/MCD43A4) provides reflectance data at 500 m spatial resolution. Datasets are produced daily based on a 16-day retrieval period from both the Terra and Aqua spacecraft, choosing the best representative pixel from the 16-day period [122,127]. The MOD09A1 V6.1 (MODIS/061/MOD09A1) product provides an estimate of the surface spectral reflectance of Terra MODIS at 500m resolution and is corrected for atmospheric conditions. In order to cover the 1984–2018 period, four collections of LANDSAT imagery were used: LANDSAT 5 MSS SRTier 1 (LANDSAT/LM05/C01/T1_SR), for 1989, LANDSAT 5 TM SR Tier 1 (LANDSAT/LT05/C01/T1_SR), for the period 1984–2011, LANDSAT

7 ETM + SR Tier 1 (LANDSAT/LE07/C01/T1_SR), for 1999–2018, and LANDSAT 8 OLI/TIRS (LANDSAT/LC08/C01/T1_SR), for 2013–2018 and MODIS for 2000–2018.

All these satellite data were downloaded for the vegetative period (from April to October) of each year. For the target period of 35 years, in the Sisian and Meghri study cases a total of 1035 and 663 satellite images were used, respectively.

### 2.2.3. NDVI Data Computation

The NDVI (1) values based on the red and near infrared bands were obtained from the MODIS and LANDSAT sensors for the two study areas corresponding to the vegetation period over 35 years (1984–2018) using a JavaScript API in GEE.

$$NDVI = \frac{(NIR - RED)}{(NIR + RED)} \qquad (1)$$

Subsequently, 10-day (MODIS) and monthly (LANDSAT) NDVI images were derived by calculating the median NDVI value of all available scenes for the indicated time-period of each year. Median NDVI is less affected by outlier values [60]. Median seasons NDVI images from 1984 to 2018 were computed by averaging the monthly NDVI images of each year [16]. Then mean NDVI values from all grassland pixels within each study site were extracted for further analysis. During the analyses we considered an important fact, that LANDSAT OLI data are reported to have some differences when compared with previous LANDSAT sensors [48,128,129]. On average, the OLI TOA reflectance is greater than the ETM+ TOA reflectance for all bands, with greatest differences in the near-infrared. Therefore, it can also affect NDVI values. However, as shown in previous studies, the difference becomes smaller as NDVI increases [130]. Nevertheless, based on the transformation functions obtained from the previous studies, which were developed using ordinary least squares regression, the values of NDVI of LANDSAT 5/7 and 8 were normalized [129,131].

### 2.2.4. Climate Data

Daily meteorological data have been obtained from Armenia State Hydromet Service for two weather stations in study areas (Figure 1). In order to study the relationship between the NDVI and climatic factors 10-day, monthly, seasonal (spring or April–May, summer or June–August and autumn or September–October)), humid period (April–June), dry period (July–August) sum precipitation and average temperature were calculated for the period from 1984 to 2018.

### 2.3. Method
### 2.3.1. Satellite Data Pre-Processing

A series of image pre-processing steps, such as atmospheric, BRDF and topographic corrections, cloud and shadow masking were carried out to study vegetation change analysis using LANDSAT satellite data [48].

#### Atmospheric Correction and Cloud Masking

Within the Google Earth Engine environment, all satellite images were corrected to SR using the LEDAPS method for LANDSAT MSS, TM, and ETM+ and the LaSRC method for LANDSAT OLI. For applying the cloud, shadow, water, and snow masking to the LANDSAT MSS/TM–ETM/OLI image collection, the necessary CFMASK algorithm, as well as a per-pixel saturation mask [132], were created by adapting the template provided by the GEE platform.

#### Topographic Correction

Topographic correction of remote sensing data is required for mountainous terrain, because it accounts for variations in reflectance due to slope, aspect, and elevation [133–135]. The code for topographic correction in GEE was developed by [136] based on the modified

sun-canopy-sensor (SCS + C) topographic correction method (2) [137]. These methods were applied for all LANDSAT images in the GEE environment.

$$L_n = L \frac{\cos \alpha \cos \theta}{\cos \iota} \quad (2)$$

where $L_n$ is the normalized reflectance, $L$ is the uncorrected reflectance, $\alpha$ is the terrain slope, $\theta$ is the solar zenith angle and $\iota$ is the angle of incident illumination.

BRDF Effects Correction

The Bidirectional Reflectance Distribution Functions (BRDF) model is applied to reduce the directional effects due to the differences in solar and view angles between LANDSAT sensors [72,77,83,131,133].

As shown in some studies, LANDSAT 5 data may result in significant reflectance and NDVI differences due only to LANDSAT 5 orbit changes [138]. Moreover, Gao [139] and Roy [82] concluded that due to BRDF effects across the LANDSAT swath the red and NIR reflectance can vary by up to 0.02 and 0.06. Such angular effects can be corrected using a BRDF model. For the correction of BRDF effects from LANDSAT data, we used a semi-physical approach BRDF normalization proposed by Roy [82]. Regarding the MODIS NBAR (Nadir BRDF Adjusted Reflectance) imagery, they are provided to GEE [140]. The implementation of BRDF correction of LANDSAT images in GEE was developed by [136] based on Roy's [82] algorithms.

2.3.2. Statistical Analysis

Numerous methods have been adopted to estimate spatiotemporal vegetation changes and relationships between climatic factors, such as correlation and regression analysis, Sen, and Mann–Kendall models [82,141,142]. These methods have been validated for vegetation variation [82,101,143]. A Mann–Kendall test (3) and Sen's slope (called Sen's slope because of set of pairs $(i, x_i)$) method (4) were used to estimate NDVI trends, and the Pearson correlation method (5) between NDVI and temperature and precipitation were computed to identify the role of climate factors in vegetation changes.

$$S = \sum_{k=2}^{n-1} \sum_{j=k+1} sign(x_j - x_k)$$
$$sign(x_j - x_k) = \begin{cases} 1 \ if \ x_j - x_k > 0 \\ 0 \ if \ x_j - x_k = 0 \\ -1 \ if \ x_j - x_k < 0 \end{cases} \quad (3)$$

where: $x_1, x_2, x_3 \ldots x_n$ represent $n$ data points, $x_j$ represents the data point at time $j$ and S is the Mann–Kendall statistic.

$$\text{Sen's slope} = \text{Median} \left\{ \frac{x_j - x_i}{j - i} \right\} : i < j \quad (4)$$

where $x_i$ is a time series.

$$r = \frac{\sum (x_i - \overline{x})(y_i - \overline{y})}{\sqrt{\sum (x_i - \overline{x})^2 \sum (y_i - \overline{y})^2}} \quad (5)$$

$$r = correlation \ coefficient$$

$$x_i = values \ of \ the \ x - variable \ in \ a \ sample$$

$$\overline{x} = mean \ of \ the \ values \ of \ the \ x - variable$$

$$y_i = values \ of \ the \ y - variable \ in \ a \ sample$$

$$\overline{y} = mean \ of \ the \ values \ of \ the \ y - variable$$

Trend Analysis

To estimate the directional changes in the NDVI time series, trend analyses were performed using data from different sensors [16,42,141]. Trend analysis was derived using the nonparametric original Mann–Kendall [142,143] and modified Mann–Kendall trend tests [144], and Sen's slope estimator [145,146], all implemented based on the python package [147]. The advantage of Mann–Kendall is its suitability for small sample sizes and robustness against the effects of outliers [148]. However, the Mann–Kendall trend test is based on uncorrelated data, and test results tend to be affected by the persistence of the time series [149,150]. In order to reduce the impact of serial correlation on the Mann–Kendall test, modified Mann–Kendall methods such as pre-whitening; free pre-whitening; Hamed and Rao; and Yue and Wang approaches [144,148–151] were used. Modified Mann–Kendall methods were applied by many studies for NDVI long-time series trend analysis [42,60].

The Kendall tau (τ) obtained from Mann–Kendell statistics was also used to measure the strength of the relationships. The Mann–Kendall tau coefficient is ranged from −1 to 1, τ = 1 means a consistently increasing trend, while τ = −1 means a consistently decreasing trend, if τ = 0 means no trend exists. Significance of trend can be evaluated by using standardized z-score or *p*-value. A z-value $\geq$ 1.96 represents a statistically significant increase, while a z-value $\leq$ −1.96 indicates a significant decrease at the 95% (α = 0.5) significance level, if z-value is above or below 2.57 and −2,57, respectively, the trend is significant at the 99% level [152,153]. In this study, to test the significance of trends, as well as significance test was applied: strong significant ($p < 0.01$); significant ($0.01 < p < 0.05$); lower significant ($0.05 < p < 0.1$); insignificant level (*p* greater than 0.1) according to the F-test [47,154].

In addition to the Mann–Kendall method, Sen's slope was also used to estimate the slope of NDVI. The Sen's slope estimation model [145] is suitable for the qualitative description of time series trends. Sen's slope is better for identifying the magnitude of the trend. If the Sen's slope result is positive, the time series has an increasing trend; when the result is negative, the time series has a decreasing trend. In the last decades, it was widely applied in the studies of vegetation dynamics [12,153].

Correlation and Time-Lag Effect Analysis

Pearson linear correlation coefficients (r) [30] were calculated to investigate temporal relationships between NDVI and precipitation and temperature across different time bases: spring, summer, autumn, dry period, humid period, months, and 10-day periods, respectively. The statistical significance of correlations was evaluated based on the t-test (at a significance level of 0.05 and 0.01). Moreover, the Pearson correlation was applied to analyze the time-lag effects between NDVI and climatic factors [23,40,46,101,104]. Considering that the influence of climate change on vegetation dynamics is a cumulative process [155], the response of vegetation growth to regular climatic factors (precipitation/temperature) in mountain ecosystems may have a time lag of 10-day, months or seasons (2–3 months) [13,19,22,40,46,113,156].

Thus, Pearson correlation analysis between 10-day, monthly, and seasonal NDVI and average 10-day, monthly, and seasonal temperature/precipitation during 1984–2018 growing seasons for two study areas were performed. All statistical correlation analysis was done using the R programming language.

## 3. Results and Discussion

### 3.1. Relationships between NDVI and Climate Variables

3.1.1. Correlation and Time-Lag Effects between MODIS NDVI Data Series and Climatic Factors

The correlations between the 10-day-averaged climate data and NDVI values were assessed using MODIS data with BRDF and topographic, atmospheric corrections as well as MODIS SR data with only atmospheric correction. The latter will give an opportunity to understand the usefulness of applied preprocessing methods for the study of relationships. As can be seen from

the tables of the two study areas (Sisian, Meghri) provided in the Supplementary Materials, a clear difference emerges between the correlation matrices with and without corrections. The correlation of climate data with the MODIS BRDF NDVI follows a regular pattern, in contrast to the MODIS SR NDVI, where the correlations seem to behave more randomly (Figure 2; Tables S1 and S2).

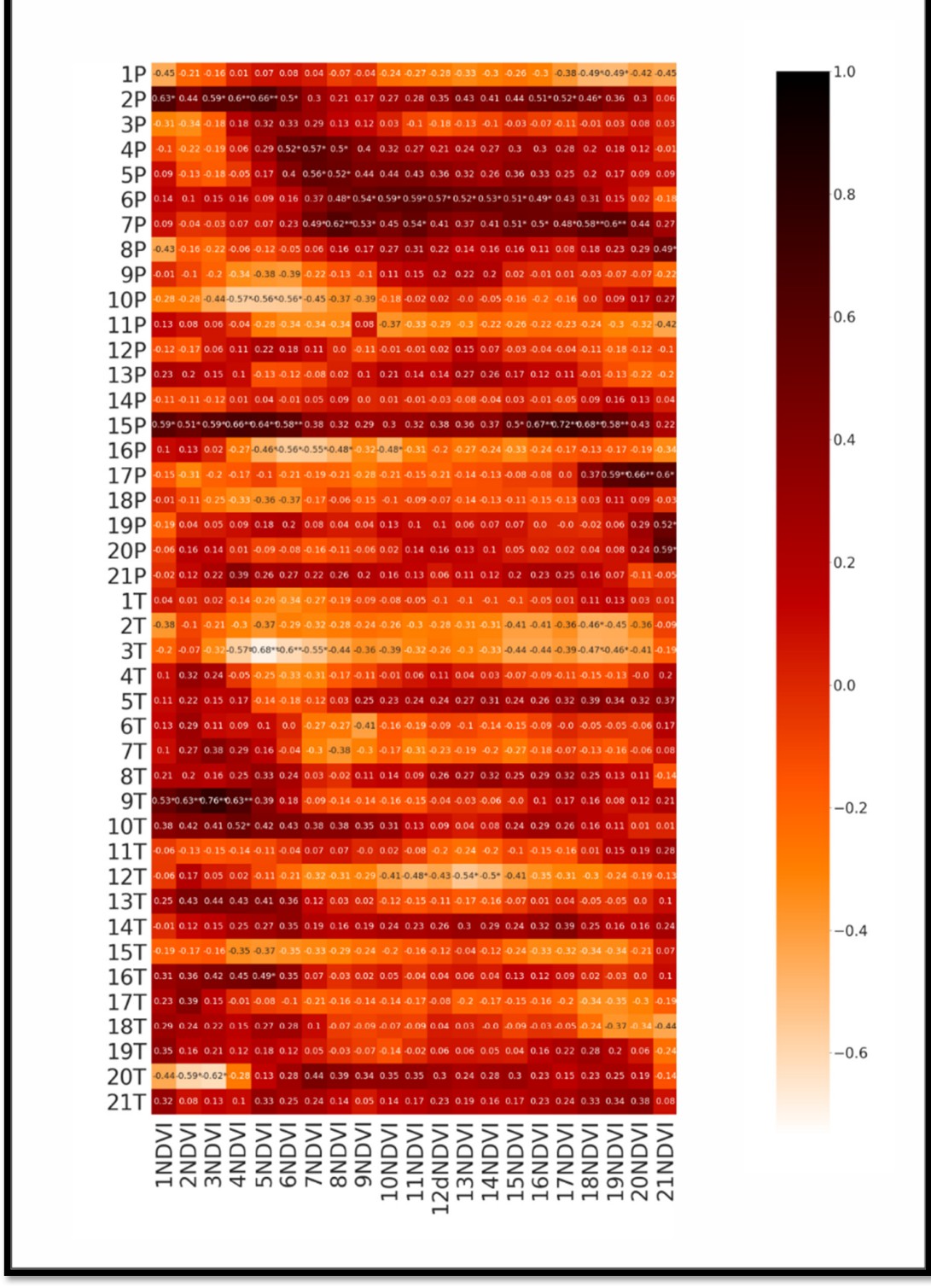

**Figure 2.** Correlation between 10-day values of temperature, precipitation and BRDF NDVI for Meghri, with the levels of significance: *—*p* < 0.05, ** *p* < 0.01.

Considering the above-mentioned factors, we decided to opt for corrected rather than uncorrected data. Thus, in case of a large number of correlations, we discuss only those correlations between MODIS BRDF NDVI and climatic factors, which are statistically significant at the 0.05 and 0.01 level.

In the case of the Meghri arid study area, the correlation between NDVI and temperature/precipitation for the same 10 days is weak (Table S1) Instead, high correlations were found between NDVIs and previous 10-day temperatures/precipitation. For instance, Table S2 shows that NDVIs in the 4th-6th and 13th-14th 10 days feature a significant negative correlation with temperature values in 3rd and 12th 10-day periods, respectively. In the case of precipitation, there is a significant time-lag effect of 2nd, 3rd, 4th, 5th, 6th, 7th, 15th, and 17th 10-day precipitation on NDVI values in 3rd–6th, 6th–8th, 7th–8th, 8th–16th, 7th–19th, 15th–19th, 19th–21st 10 days, respectively.

For the Sisian semi-arid area, an obvious time-lag effect of climatic factors on vegetation was observed, too (Figure 3, Tables S3 and S4). However, in contrast to Meghri, significant correlation was also observed for the same 10 days. For instance, NDVIs in the 4th, 5th, 7th, 12th, and 17th 10 days feature a significant positive correlation with precipitation in the same 10 days whereas NDVIs in the 12th, 13th, 16th, and 18th 10 days feature a significant negative correlation with temperature in the same 10-day periods. Precipitation in the 6th–8th, 10th, 12th–13th, and 17th 10 days have a significant time-lag effect on the 8th–11th, 8th–11th, 9th, 12th–13th, 13th–19th, 15th–19th, and 18th–21st 10-day NDVIs, respectively. Temperature also had a significant negative time-lag effect on NDVI, particularly in the case of the 2nd, 7th, 12th–13th, and 16th 10 days.

In summary, we may conclude that, in general, precipitation for the Meghri arid area had a time-lag effect, which started from the first 10 days, in contrast to the Sisian semi-arid area, where the time-lag effect was observed only from the sixth 10-days. The correlation between NDVI and temperature for the Meghri case is weak. However, the 6th, 7th, 12th, and 17th 10 days have a significant time-lag effect on vegetation for both areas.

### 3.1.2. Correlation and Time-Lag Effects between LANDSAT NDVI Data Series and Climatic Factors

To study relationships between climate factors and NDVI values, long time series (1984–2018) of LANDSAT data were considered. As the frequency of data acquisition for the LANDSAT sensor is lower (2–8 images per month), in this case the average monthly NDVI values were calculated. The response of NDVI to climate factors may differ according to vegetation type and growth phase [157] with obvious time-lag effects [13,22,104]. In some regions, NDVI variation is due to variation in precipitation [25,66,102,158–160]; in other regions, temperature is a major influencing factor in the variation of NDVI [34,46,92,93]. However, for the main cases of NDVI changes are dependent on both temperature and precipitation climatic factors [30,93,161,162]. Therefore, to recognize seasonal differences and lag effects of climatic factors on NDVI within the growing season, we performed correlation analyses between seasonal mean NDVI and precipitation, temperature. The correlations were assessed using LANDSAT data with BRDF and topographic, atmospheric corrections as well as LANDSAT SR data with only atmospheric correction. As a result, for the two study areas (Sisian, Meghri), there is a clear similarity between those two correlation matrices. Only a slight difference between the Meghri area NDVI and temperature correlation matrices was observed. Therefore, it seems that both the LANDSAT BRDF-Adjusted Reflectance and LANDSAT SR data can be used to study the regularities of time-lag effect of monthly and seasonal precipitation/temperature on NDVI.

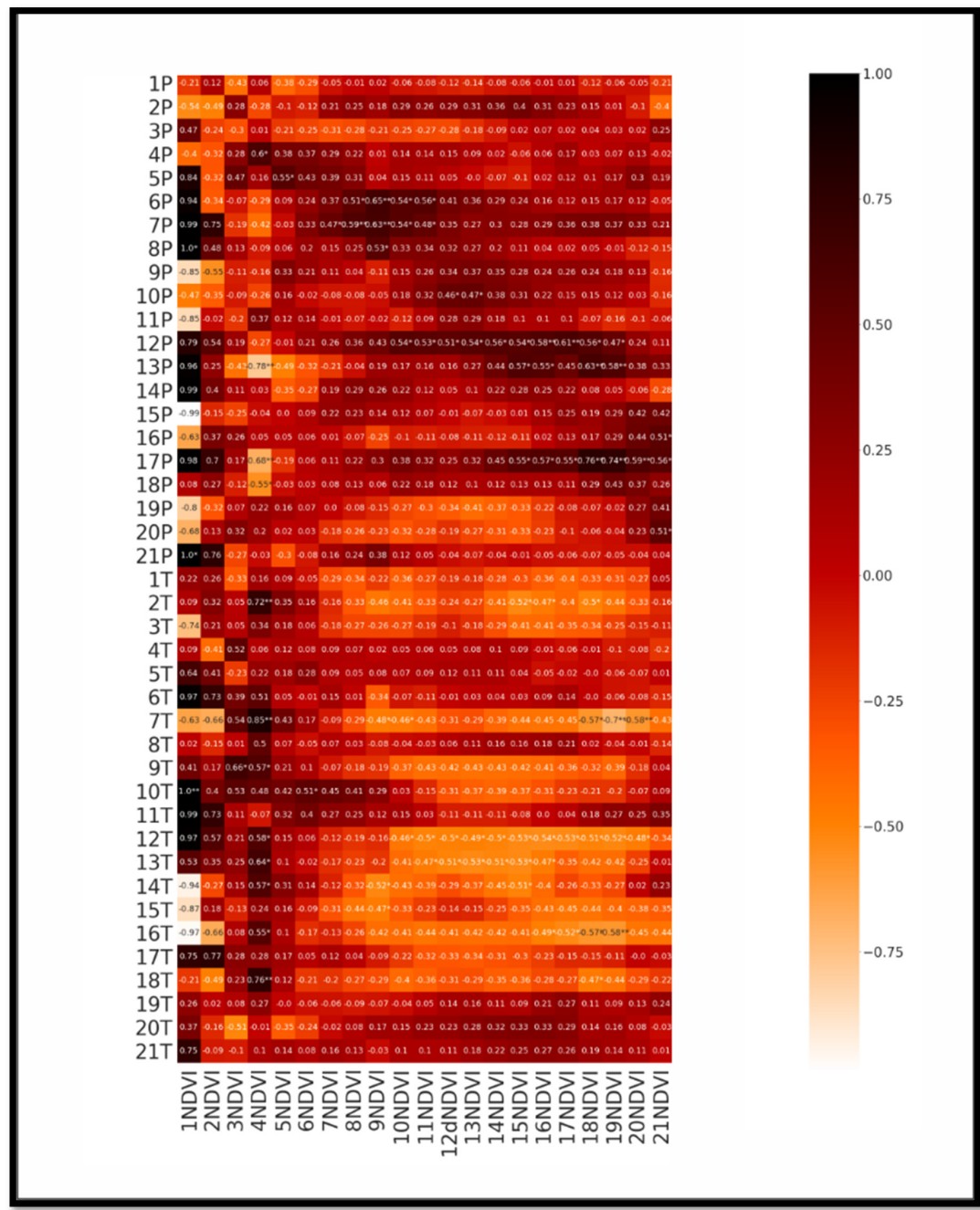

**Figure 3.** Correlation between 10-day values of temperature, precipitation and BRDF NDVI for Sisian, with the levels of significance: *—$p < 0.05$, ** $p < 0.01$.

Figure 4, Tables S5 and S6 show that for the Meghri study area, summer and autumn NDVI feature significant positive correlation with spring and summer precipitation, respectively. Dry period NDVI significantly correlates with humid period precipitation and dry period temperature. For the Meghri arid area, a strong time-lag effect of climatic factors on vegetation was observed. In particular, the NDVIs of June-September, September, and October significantly positively correlate with precipitation in May-June, August, and September. For the Sisian semi-arid study area (Figure 5, Tables S7 and S8), summer and autumn NDVIs correlated positively with summer precipitation and negatively with summer temperature. Dry period NDVI significantly correlated with both humid and dry period precipitation and temperature. As in the case of Meghri, in the Sisian study the NDVI of June-September significantly correlated with precipitation in May-June. However, correla-

tion between NDVI and temperature is higher than in Meghri. There exists a significant time-lag effect of April-August temperature on June, July, August, and September NDVIs. The impact of the current month temperature on the NDVI was obvious in the case of April, August, and September. In general, for arid and semi-arid study areas, NDVI correlated positively with precipitation and negatively with temperature. Meanwhile, the correlation between NDVI and precipitation was significant and showed strong lag effects.

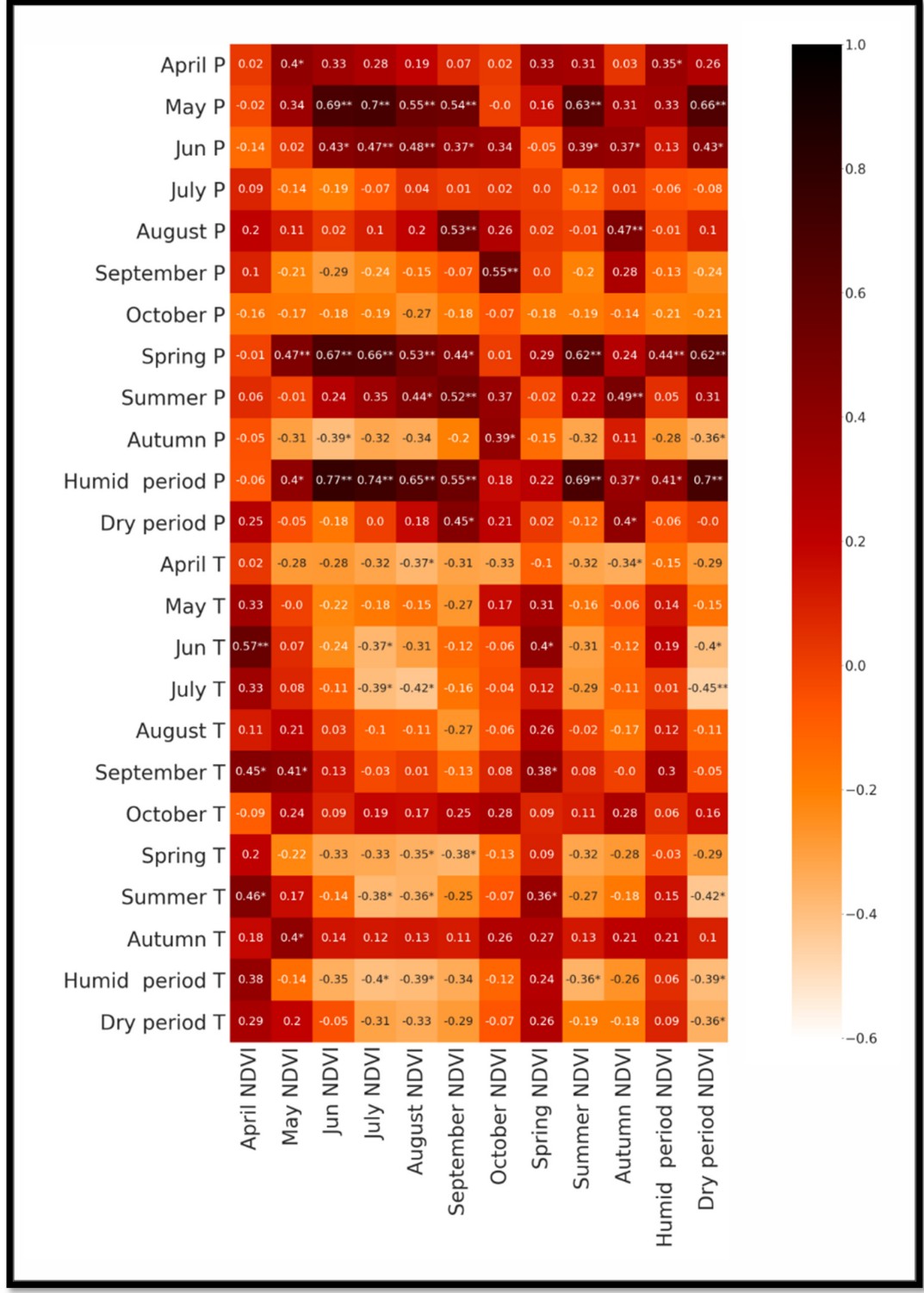

**Figure 4.** Correlation between averaged monthly and periodic values of temperature, precipitation, and BRDF NDVI for Meghri, with the levels of significance: *—*p* < 0.05, ** *p* < 0.01.

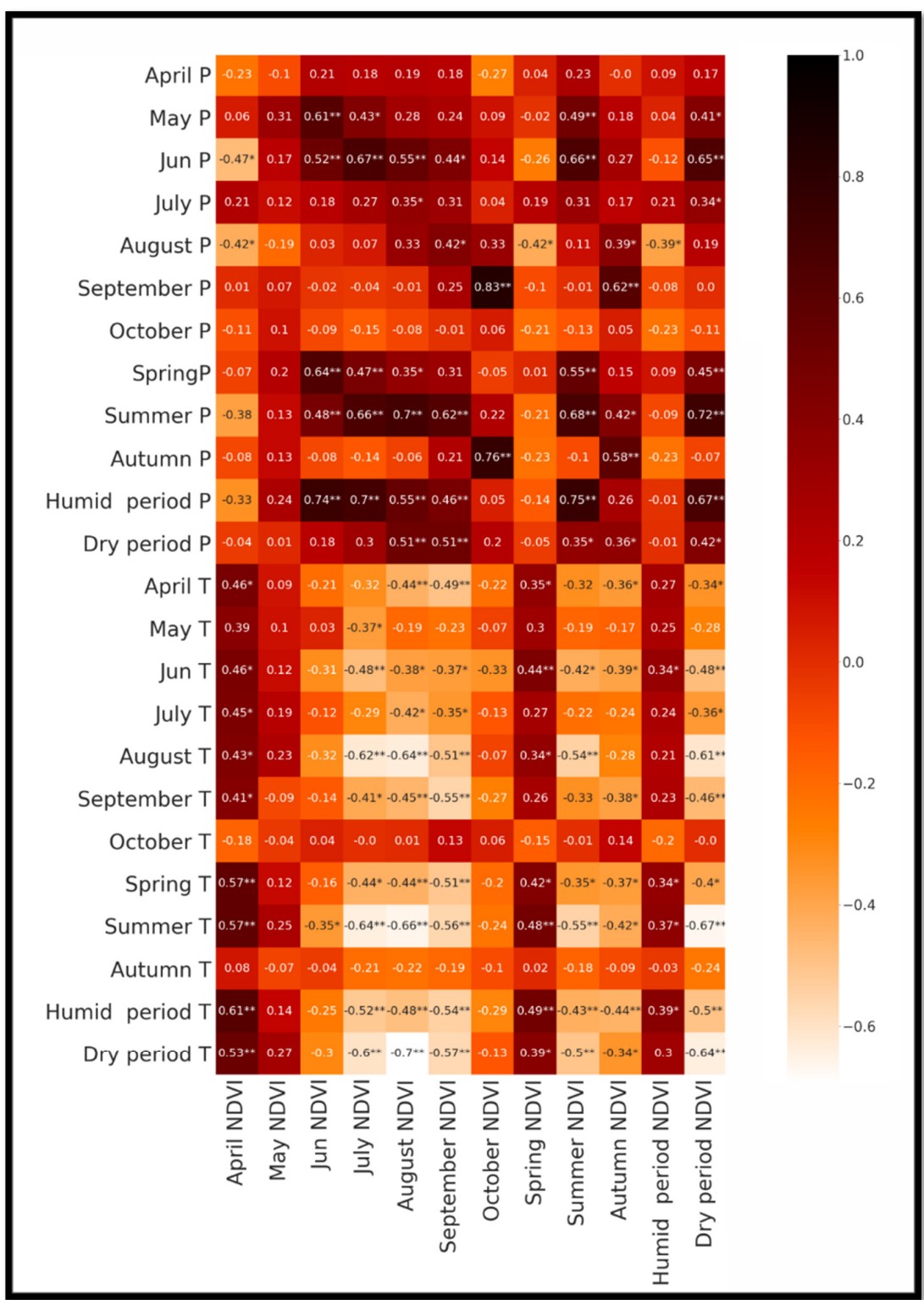

**Figure 5.** Correlation between averaged monthly and periodic values of temperature, precipitation, and BRDF NDVI for Sisian, with the levels of significance: *—*p* < 0.05, ** *p* < 0.01.

### 3.1.3. Comparison of LANDSAT- and MODIS-Based Products

LANDSAT and MODIS satellite images are used in various studies to analyze the relationship between vegetation of grasslands and climatic factors, as well as the time-lag effect. Therefore, it is necessary to compare the results obtained through those two sensors. For this purpose, the average monthly and seasonal LANDSAT/MODIS NDVI data (with

BRDF and topographic, atmospheric correction) were calculated for the 2000–2018 period (Figures 6–9). The period 2000–2018's monthly data were selected to provide comparability of MODIS and LANDSAT data.

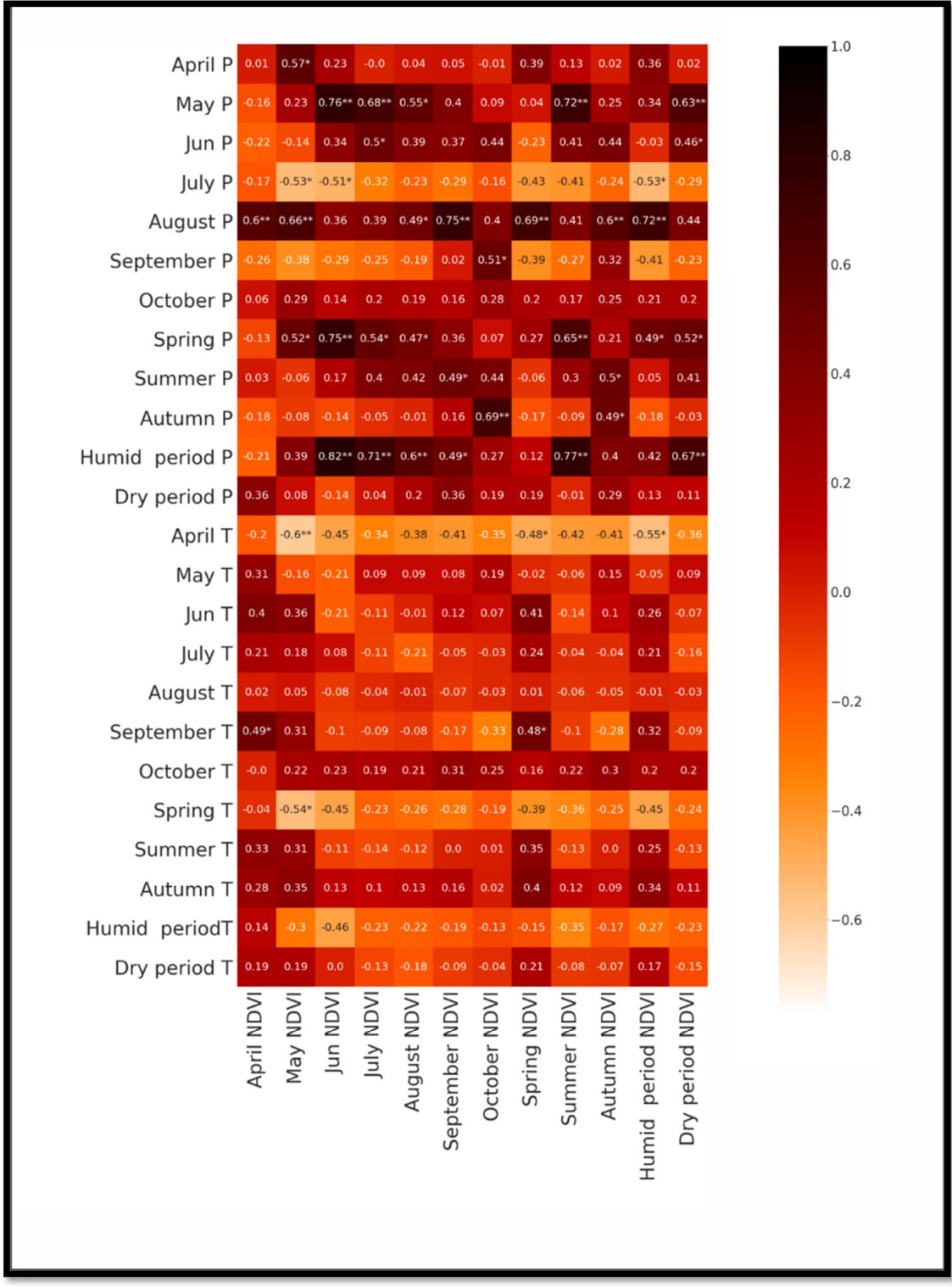

**Figure 6.** Correlation between temperature, precipitation, and the average monthly, seasonal MODIS NDVI data (with BRDF and topographic, atmospheric correction) for Meghri for the 2000–2018 period, with the levels of significance: *—$p < 0.05$, ** $p < 0.01$.

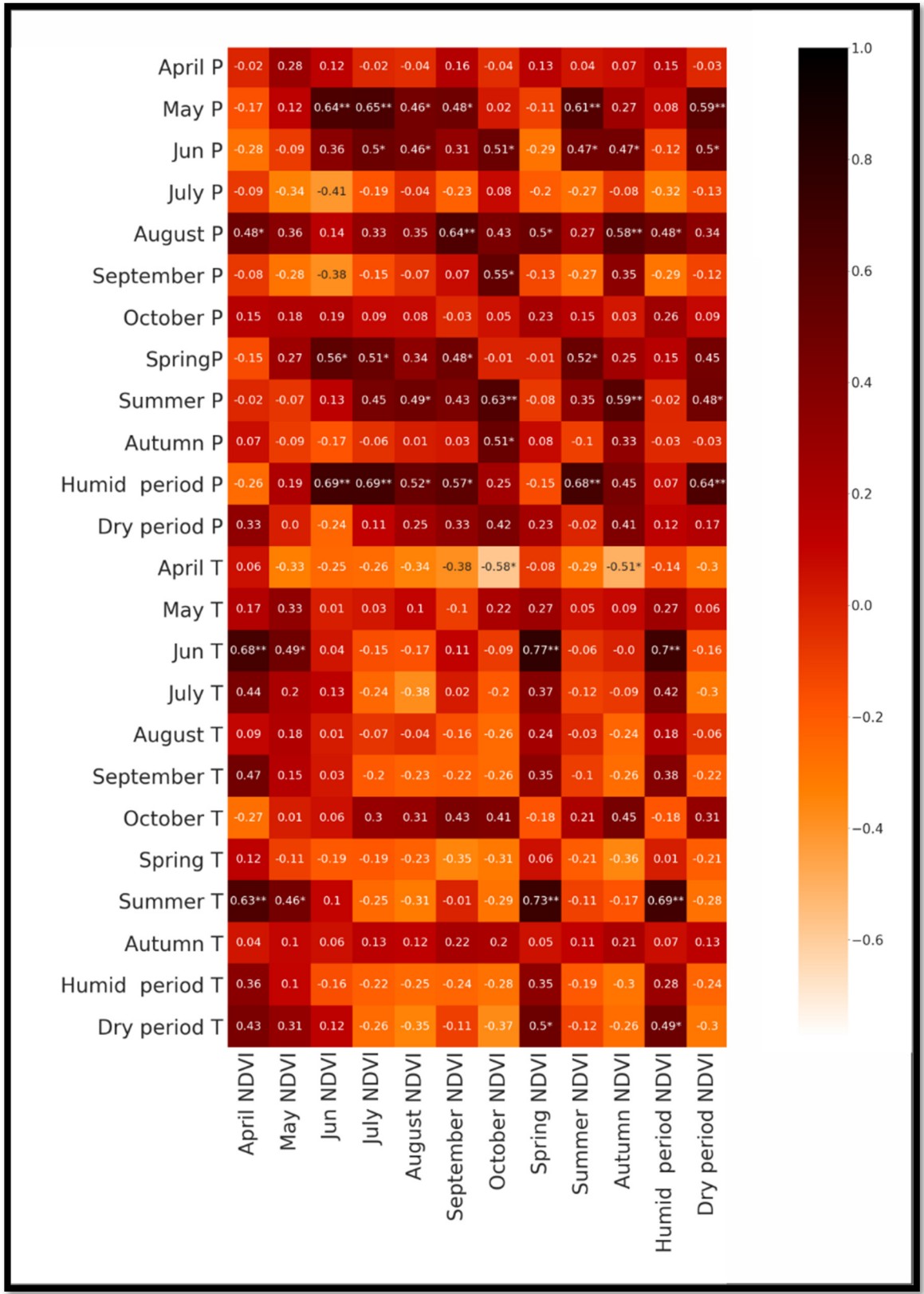

**Figure 7.** Correlation between temperature, precipitation, and the average monthly, seasonal LANDSAT NDVI data (with BRDF and topographic, atmospheric correction) for Meghri for the 2000–2018 period, with the levels of significance: *—$p < 0.05$, ** $p < 0.01$.

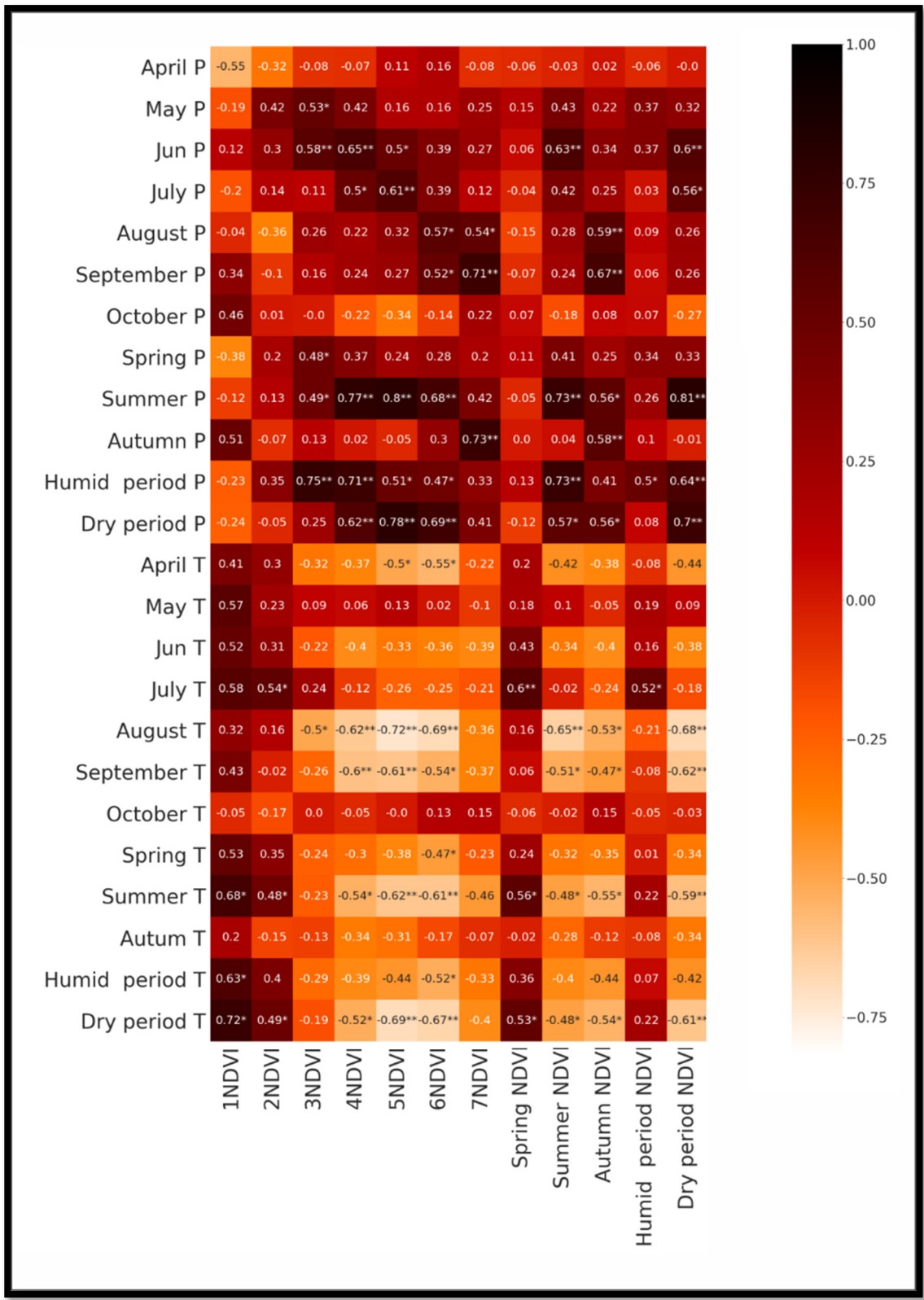

**Figure 8.** Correlation between temperature, precipitation, and the average monthly, seasonal MODIS NDVI data (with BRDF and topographic, atmospheric correction) for Sisian for the 2000–2018 period, with the levels of significance: *—*p* < 0.05, ** *p* < 0.01.

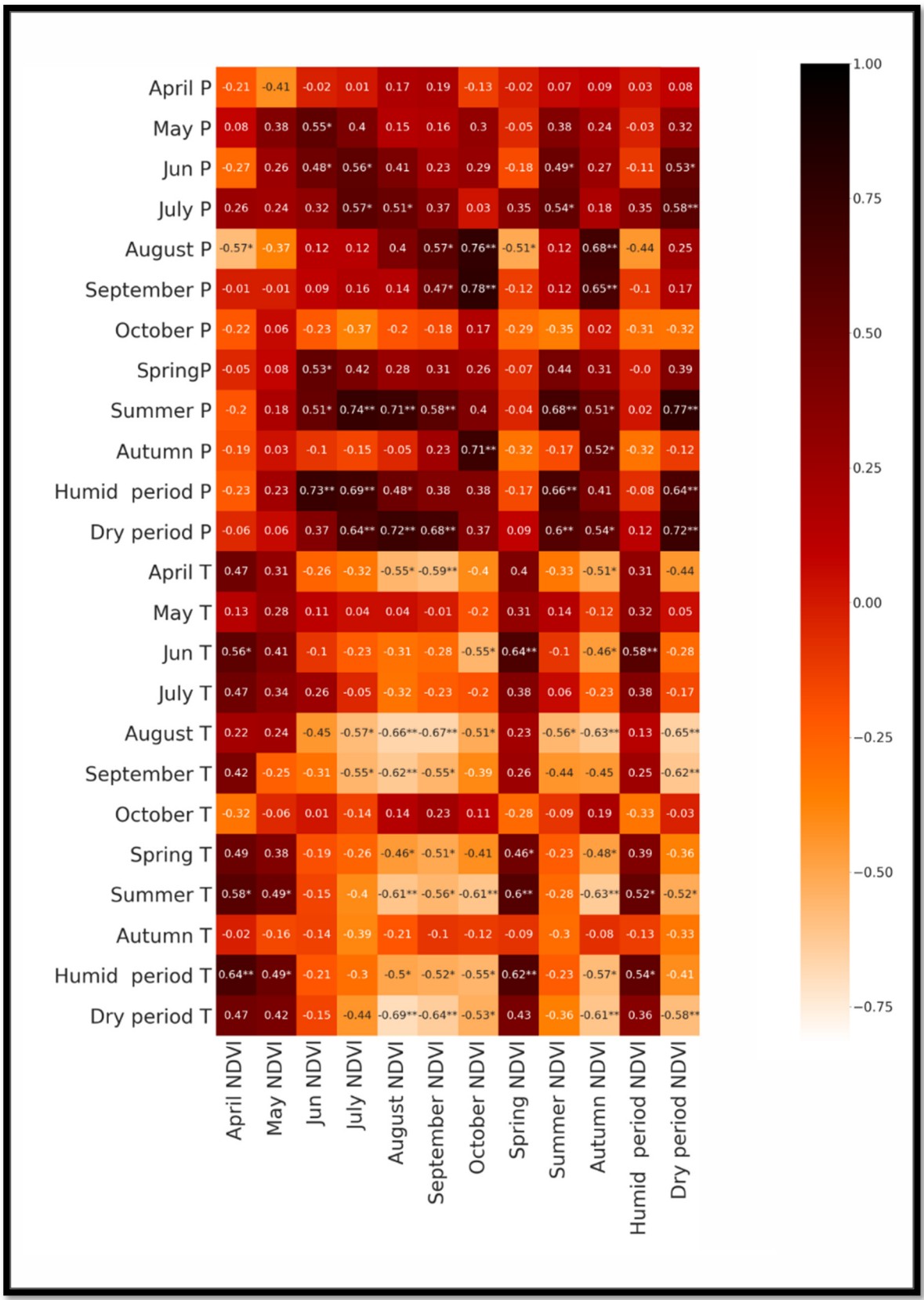

**Figure 9.** Correlation between temperature, precipitation, and the average monthly, seasonal LANDSAT NDVI data (with BRDF and topographic, atmospheric correction) for Sisian for the 2000–2018 period, with the levels of significance: *—$p < 0.05$, ** $p < 0.01$.

As seen from the results, correlation matrices are similar in the two study areas. However, there are also some nuances; for instance, in the case of Meghri, strong correlation between LANDSAT NDVI (July, August) and June precipitation was observed, but MODIS NDVI significantly correlated with June precipitation only in July. For the Meghri arid area, correlation between both MODIS, LANDSAT NDVI and temperature is weak. For semi-arid grassland, the correlation matrices in case of precipitation are identical. A significant difference is observed only in the case of spring and summer temperature.

Summarizing the above-mentioned results, we can conclude that during study of relationships between NDVI and climatic factors for the 2000–2018 period in small grassland areas, both MODIS and LANDSAT BRDF-Adjusted satellite products were suitable.

*3.2. Spatiotemporal Analyses of Trend Using LANDSAT NDVI Time Series*

To analyze long-time spatiotemporal changes in vegetation of mountain grassland, LAND-SAT NDVI were used, which provide a long-time data series (1984–2018). Mann–Kendall (MK) and four modified Mann–Kendall methods (MMK) were applied to examine the trends. Previous studies showed better performance of the MMK method with the monthly data [163]. The results of the MK and MMK tests applied to the NDVI data series at a confidence interval of 90% are shown in Table S9. In an experiment reported in the scientific literature, the trend is considered significant only if at least three of the five tests suggest a significant trend [164]. At the same time, Sen's slope method was applied to discover trends in time series. The results were divided into a stable or no-trend region ($-0.0005 <$ and $\leq 0.0005$), an increasing region ($>0.0005$), and a decreasing region ($\leq 0.0005$) [153].

Statistical results are shown in Table S9. Trend analysis indicated both significant increase and no significant changes in monthly NDVI. In general, the results of the test indicated a strong significant increasing trend in April and May NDVIs at a confidence interval of 99% for the Meghri arid area (0.003, 0.002 year$^{-1}$; respectively) and a no-trend or insignificant increasing/decreasing in all time periods NDVI at the Sisian semi-arid area. The MMK Yue and Wang test result at the Sisian area indicated a decreasing NDVI trend in September at a confidence interval of 95% ($-0.0005$ year$^{-1}$), while the other Mann–Kendall tests did not indicate any significant trend. Also, a significant increase in the mean spring, humid period NDVI during 1984–2018 was observed for the Meghri arid area (Sen's slope = 0.002, 0.001; respectively). Trend analysis of monthly precipitation indicated no trend or insignificant changes at both the Sisian and Meghri study areas. In contrast, the significant increase of temperature during 1984–2018 was detected. It is interesting to note that the modified Mann–Kendall methods suggested by Yue and Wang [148] showed a significant trend in most of the cases as compared with other methods. The same situation was observed also in other investigations [164].

During the growing season, the correlation between NDVI and temperature, precipitation was negative and positive, respectively, indicating the positive precipitation effects on vegetation growth, and negative temperature effect, despite some shifts that were mainly insignificant. However, as shown in Figures 2, 4, 6 and 8, correlation between temperature and NDVI is lower than correlation between precipitation and NDVI, particularly in the Meghri arid study area, which was justified by some studies [58,102,162,165]. It suggests that increasing temperature would not promote a decrease in vegetation. Considering the fact that precipitation has no trend, it can be justified that the NDVI of the two study areas was not changed in all months during 1984–2018, except for the spring months in the Meghri arid area, where a significant increasing trend was recorded. The latter cannot be explained precisely by the available data; therefore, it is necessary to carry out additional analyses that include climatic factors of the previous months (February-March). Nevertheless, NDVI increased during spring and was stable during autumn, suggesting that the growing season may be starting earlier [12]. However, in general, no significant change in grassland vegetation has been recorded in the study area over the last three decades. In contrast, long-term positive changes in grassland vegetation (32.7% of grasslands) for the same period were found in the Caucasus region [58].

## 4. Conclusions

This research analyzed temporal trends of MODIS and LANDSAT NDVI climatic factors, and studied the correlations and time-lag effects between grassland vegetation and climatic factors during the growing seasons (April–October) of the period 1984–2018 in Syunik region. For this purpose, 10-day, monthly, and seasonal NDVI and precipitation, temperature data for two different study areas were used.

The results show that the correlation matrix of climate data with the MODIS BRDF NDVI follows a regular pattern, in contrast to the MODIS SR NDVI, where the correlations seem to behave more randomly. The results also confirm that for the Sisian and Meghri study areas, there is a clear similarity between two correlation matrices: climatic data with LANDSAT BRDF; TC; AC NDVI, and climatic data with LANDSAT SR NDVI. It seems that both the LANDSAT BRDF:TC; AC; and LANDSAT SR data can be used to study the regularities of time-lag effect of monthly and seasonal precipitation/temperature on NDVI.

The correlation between NDVI and climatic factors show that temperature had a negative impact and precipitation had a positive impact on vegetation growth in both arid and semi-arid areas. Meanwhile, the correlation between NDVI and precipitation was significant and has an apparent time-lag effect, yet a suitable time frame has to be selected for the phenomenon to become visible. In mountain arid and semi-arid grasslands, 10-day data are more suitable for understanding the impact and time-lag effect of climatic factors on vegetation growth. For instance, the precipitation for the Meghri arid area had a time-lag effect, which started from the first 10 days, in contrast to the Sisian semi-arid area, where the time-lag effect was observed only from the sixth 10 days. The correlation between NDVI and temperature for the Meghri case is weak. However, the 6th, 7th, 12th, and 17th 10 days have a significant time-lag effect on vegetation for both areas.

Finally, we can conclude that for studying relationships between NDVI and climatic factors for the 2000–2018 period in grassland at a very local scale, both MODIS and LANDSAT BRDF-adjusted satellite products are suitable.

In general, the analysis of changes of vegetation data over 45 years (1984–2018) of mountain grasslands showed no significant change in NDVI. However, other, more specific patterns have also been observed in the data. For example, a strong significant increasing trend in the April and May NDVIs at a confidence interval of 99% in the Meghri arid area (0.003, 0.002 year$^{-1}$; respectively) and an insignificant increasing/decreasing in all time periods' NDVIs in the Sisian semi-arid area. The MMK Yue and Wang test result in the Sisian area indicated a decreasing NDVI trend in September at a confidence interval of 95% ($-0.0005$ year$^{-1}$), while the other Mann–Kendall tests did not indicate any significant trend.

There are however other factors to consider, which will be incorporated in future phases of our research. Last but not least, the effect of human activity on grasslands was not considered, although it surely has an impact on the evaluated variables. To refine the study, we will have to determine how to effectively remove anthropogenic impact, in order to establish a "cleaner" quantitative correlation between climatic factors and vegetation.

These results can assist to better understand the relations between NDVI and climatic factors and may provide guidance for the sustainable management of the mountain's grasslands by decision makers.

**Supplementary Materials:** The following supporting information is provided as a .docx file (tables.docx), which contains nine tables as follows and can be downloaded at: https://www.mdpi.com/article/10.3390/geosciences12110412/s1, Table S1: The correlation between 10-day values of precipitation and SR NDVI for Meghri (significance level: *—$p < 0.05$, ** $p < 0.01$); Table S2: The correlation between 10-day values of temperature and SR NDVI for Meghri (significance level: *—$p < 0.05$, ** $p < 0.01$); Table S3: The correlation between 10-day values of precipitation and SR NDVI for Sisian (significance level: *—$p < 0.05$, ** $p < 0.01$); Table S4: The correlation between 10-day values of temperature and SR NDVI for Sisian (significance level: *—$p < 0.05$, ** $p < 0.01$); Table S5: Correlation between averaged monthly and periodic values of precipitation and SR NDVI for Meghri, with the levels of significance: *—$p < 0.05$, ** $p < 0.01$; Table S6: Correlation between averaged monthly and periodic values of temperature and SR NDVI for Meghri, with the levels of

significance: *—$p < 0.05$, ** $p < 0.01$; Table S7: Correlation between averaged monthly and periodic values of precipitation and SR NDVI for Sisian, with the levels of significance: *—$p < 0.05$, ** $p < 0.01$; Table S8: Correlation between averaged monthly and periodic values of temperature and SR NDVI for Meghri, with the levels of significance: *—$p < 0.05$, ** $p < 0.01$; Table S9: Trend analysis based on LANDSAT NDVI time series.

**Author Contributions:** Conceptualization, V.M., S.A. and F.D. Methodology, V.M. and F.D. Data analysis and validation, V.M. and F.D. Writing—original draft preparation, V.M., G.A. and F.D. Writing—review and editing, V.M., S.A., G.A. and F.D. Visualization, V.M., G.A. and F.D. All authors have read and agreed to the published version of the manuscript.

**Funding:** This work was supported by the Science Committee of the Republic of Armenia, in the framework of research project ML4GEO (funding number: No. 20TTCG-1E009). The project timeframe is 2020–2023.

**Institutional Review Board Statement:** Not applicable.

**Informed Consent Statement:** Not applicable.

**Data Availability Statement:** Not applicable.

**Acknowledgments:** The authors wish to thank the University of Pavia for awarding a visiting grant to Vahagn Muradyan in the framework of the CICOPS initiative. His visiting professor time in Pavia was extremely useful in advancing the work described in this paper.

**Conflicts of Interest:** The authors declare no conflict of interest. The funders had no role in the design of the study; in the collection, analyses, or interpretation of data; in the writing of the manuscript; or in the decision to publish the results.

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
