# Peer review of "Multidecadal Trend Analysis of Armenian Mountainous Grassland and Its Relationship to Climate Change Using Multi-Sensor NDVI Time-Series"

_geosciences, doi:10.3390/geosciences12110412_

Round 1
Author Response
See attached PDF

Reviewer 2 Report
The manuscript addresses important and timely issues related to mountainous grassland and its relationship to climate change.
Please limit the number of Tables and Figures to a minimum.
Despite an extensive literature review, References contains few items from 2022.
The authors of the article point out that "Results suggest that temperature and precipitation had negative and positive impacts on vegetation growth, respectively, in both areas". Was it possible to eliminate other factors that may have influenced the growth of vegetation, e.g. soil quality?
Some excerpts from the chapter Conclusions may need to be clarified, as they may seem too obvious, e.g. "Results suggest that temperature and precipitation had negative and positive impacts on vegetation growth, respectively, in both areas".
Reviewer 3 Report
Dear Authors,
I'll start my analysis congratulating you for this interesting article and for your hard work.
There are not many things that I can add, in order to improve the submitted article. In my humble opinion, it's almost perfect as it is.
My observations are:
1. Please choose a smaller font for Tables or "AutoFit to Window" function, in order to be fully visible in the printed page.
2. Refer to Figures with "Figure(s) X", instead of "Fig. ..." or "Figs. ...".
3. Pay attention to measure units, your superscripts are missing (km2, year-1 etc.).
4. Please verify again the way of citing papers - they are sometimes mentioned one by one ([3], [4], [5], [6]), other times concentrated ([21]–[36]). Also, it seems that "References" are written in another format (font, size, line spacing).
The content is accurate, with a lot of original data, I cannot make any suggestions.
Good luck in you research work!
Best wishes,
M. Berca
Round 2
Reviewer 1 Report
The article has been modified accordingly to reviews and the description of the research is clearer.
Author Response
The authors wish to thank the reviewer for his/her positive assessment of the amendments made to the manuscript. We are grateful for the suggestions which helped make the content clearer and more useful.
The manuscript file has now been cleaned of change markers and uploaded.
Reviewer 2 Report
The revised manuscript may be accepted for publication.
Author Response
The authors wish to thank the reviewer for his/her positive assessment of the work we have made on the manuscript to improve it according to the remarks made.
We are grateful for the suggestions which helped make the content clearer and more useful.
The manuscript file has now been cleaned of change markers and uploaded.